

# Progress in applying CRISPR in pathogen detection

Yan Wu[1,2,3], Jimin Li[1,2], Rui Wang[4], Fengling Qiao[1,2], Jinlin Guo[1,2] and Xu Jia[3,5]

[1] College of Medical Technology, Chengdu University of Traditional Chinese Medicine, Chengdu, China
[2] Chongqing Key Laboratory of Sichuan-Chongqing Co-construction for Diagnosis and Treatment of Infectious Diseases Integrated Traditional Chinese and Western Medicine, Chengdu, China
[3] Non-Coding RNA and Drug Discovery Key Laboratory of Sichuan Province, Chengdu Medical College, Chengdu, China
[4] Institute of Microbiology, Chinese Academy of Sciences, Chengdu, China
[5] School of Basic Medical Sciences, Chengdu Medical College, Chengdu, China

## ABSTRACT

Infectious diseases caused by various pathogens are among the major threats to human health, and the key to controlling these diseases is the early diagnosis of pathogens. Currently, standard methods for clinical pathogen diagnosis have issues such as high cost, long processing time, and limited sensitivity, which are difficult to overcome with existing technology. In recent years, clustered regularly interspaced short palindromic repeats (CRISPR) gene editing technology, which utilizes a programmable endonuclease-based gene editing system, has been widely applied in the fields of treatment and diagnosis. In pathogen detection, CRISPR technology offers the advantages of being fast, accurate, sensitive, and simple, enabling the detection of various pathogens early and instantly, thereby compensating for the shortcomings of existing nucleic acid detection methods. Moreover, the precise identification and characterization of mutant genes that cause virulence and drug resistance *via* CRISPR has further promoted its application in clinical pathogen diagnosis, providing a basis for controlling pathogen transmission and monitoring resistance. Currently, although the CRISPR/Cas system offers various advantages, there are still areas for improvement in clinical applications, including cumbersome operational processes, difficulty in achieving accurate quantitative, multiplex, and standardized detection, and reliance on specialized instruments. Therefore, continuous improvement is necessary to develop new and more convenient CRISPR-based tools for pathogen detection. This review focuses on various simplified strategies of the latest CRISPR diagnostic tools, including extraction-free, amplification-free, and integrated reactions, as well as sensitive and portable output strategies, to overcome these obstacles in clinical applications and propose the next strategic direction for providing researchers with innovative strategies for real-time pathogen diagnosis.

Corresponding authors
Fengling Qiao,
qiaozhaoyi@cdutcm.edu.cn
Xu Jia, jiaxucmc@qq.com

## SURVEY METHODOLOGY

The research goal of this study was to design new and more convenient CRISPR diagnostic tools for the early diagnosis of pathogens. A total of 2,760 relevant literatures were retrieved from the PubMed electronic database, with a retrieval time set from 2012 to 2025. Keywords: CRISPR technology, CRISPR diagnosis. The first aspect of the inclusion criteria involves clinical trials and reviews with complete structures and sufficient materials, while the other aspect involves retrieving relevant keywords. Articles that were not peer-reviewed and were irrelevant to the research topic were excluded.

## INTRODUCTION

Infectious illnesses are among the most serious hazards, causing major illness and death in humans (*Hwang, Hwang & Bueno, 2018*). Early diagnosis is crucial for preventing the spread of the disease, providing timely treatment, and reducing the risk to patients and healthcare workers, especially for new or recurrent infections caused by pathogens. (*Kostyusheva et al., 2022*). Currently, routine methods for detecting pathogens include isolation culture, immunological detection, and nucleic acid detection. Among these methods, isolation cultures are time-consuming and laborious and are limited to culturable bacteria (*Scheler, Glynn & Kurg, 2014*). Immunological detection has low sensitivity, requires a long detection window, and is easily missed. As a mature detection method, nucleic acid detection has the advantages of high sensitivity and strong specificity. Polymerase chain reaction (PCR), a conventional method of nucleic acid detection, has been widely used in the field of pathogen detection (*Liu et al., 2017*; *Qiu et al., 2020*). In particular, the development of real-time quantitative PCR (qPCR) and digital PCR (dPCR) has pushed molecular diagnosis to a more sensitive and quantitative stage. However, despite its great sensitivity and specificity, PCR technology requires a specialized laboratory environment, trained workers, and well-purified materials, is expensive and time-consuming, and is limited in some fields. In particular, the emergence and spread of SARS-CoV-2 have heightened our awareness of the limitations of traditional PCR techniques (*Artesi et al., 2020*; *Zhang et al., 2022*). Therefore, nucleic acid detection techniques that are easy to use, inexpensive, time-consuming, accurate, and sensitive are urgently needed.

In recent years, the clustered regularly interspaced short palindromic repeats (CRISPR)/Cas system, which consists of clustered regularly interspaced short palindromic repeats and their associated proteins (Cas), has been developed and evolved into a novel molecular diagnostic tool, overcoming the limitations of traditional PCR and enabling rapid, efficient, low-cost and portable detection of pathogens and their mutations (*Rahimi et al., 2021*; *Zavvar et al., 2022*). The system utilizes guide RNA (gRNA) sequences encoded in the CRISPR motif to direct Cas proteins to recognize and cleave specific exogenous nucleic acid sequences. When the target nucleic acid sequence is present, the gRNA binds to the target sequence through base-pairing complementarity, which subsequently activates the Cas protein to cleave nearby single-stranded DNA or RNA, thereby amplifying the signal. Therefore, rapid CRISPR/Cas detection can be achieved by designing guide RNAs

(gRNAs) that target different genes. However, CRISPR/Cas systems alone often struggle to achieve highly sensitive pathogen detection and require a combination with nucleic acid amplification technology or different biosensors to achieve signal amplification. *Pardee et al. (2016)* developed NASBA-CRISPR (NASBACC), a CRISPR/Cas9-based nucleic acid detection platform, by combining nucleic acid sequence-based amplification (NASBA) and double-stranded DNA cleavage by Cas9 to detect Zika virus with high sensitivity. Since the discovery of the *trans*-cleavage activity of Cas13 and Cas12, CRISPR diagnostic tools based on their *trans*-cleavage ability have been developed. For example, the Specific High-Sensitivity Enzymatic Reporter Unlocking (SHERLOCK) platform combines the CRISPR/Cas13a system with recombinase polymerase amplification (RPA) (*Gootenberg et al., 2017*). The amplification product binds complementarily to the crRNA and activates the *trans*-cleavage activity of Cas13a, thereby cleaving the fluorescently labeled ssRNA probes in the reaction system and generating many detectable fluorescent signals with a limit of detection (LOD) of aM for RNA viruses. The DNA endonuclease-targeted CRISPR transporter (DETECTR) platform replaces Cas13a in the SHERLOCK platform with Cas12a for detecting DNA viruses (*Chen et al., 2018*). During the COVID-19 outbreak, the SHERLOCK and DETECTR platforms were approved by the U.S. Food and Drug Administration (FDA), representing important milestones for the use of CRISPR technology as a diagnostic tool (*Perez-Lopez & Mir, 2021*). However, although these mature CRISPR diagnostic tools have achieved sufficient sensitivity, their cumbersome operation procedures and reliance on amplification technology further hinder their clinical application and development in remote areas. Therefore, streamlining the operation process, including sample preparation, nucleic acid amplification, signal readout, and one-step and integrated reactions, to achieve faster quantitative and multiplexed assays is crucial for its further development.

Currently, most reviews on CRISPR pathogen detection focus on the applications of different types of CRISPR/Cas systems for detecting various pathogens. For example, *Wang, Shang & Huang (2020)* summarized CRISPR detection based on DNA-specific cleavage and *trans*-cleavage. *Kostyusheva et al. (2022)* described the mechanisms and functions of different classes and types of CRISPR-Cas systems, as well as the application of each type in detecting various infectious pathogens. *Fapohunda et al. (2022)* emphasized the use of the CRISPR system as a tool for diagnostics and nucleic acid detection, as well as the advantages of this powerful tool over other amplification methods. Additionally, emerging diagnostic strategies based on CRISPR, such as amplification-free assays and field applications, are being explored (*Li et al., 2023a*; *van Dongen et al., 2020*). However, no reviews have focused on simplified strategies for CRISPR pathogen diagnosis. In this work, we first review the cleavage properties of the CRISPR/Cas system and the prospects of CRISPR technology in pathogen mutation detection and then focus on various simplified strategies of the latest CRISPR diagnostic tools and their advantages and disadvantages, including extraction-free, amplification-free, and integration reactions and sensitive portable output strategies, and discuss the barriers to further clinical applications and future strategic directions, with a view to providing researchers with innovative strategies for pathogen detection and improved clinical applicability.

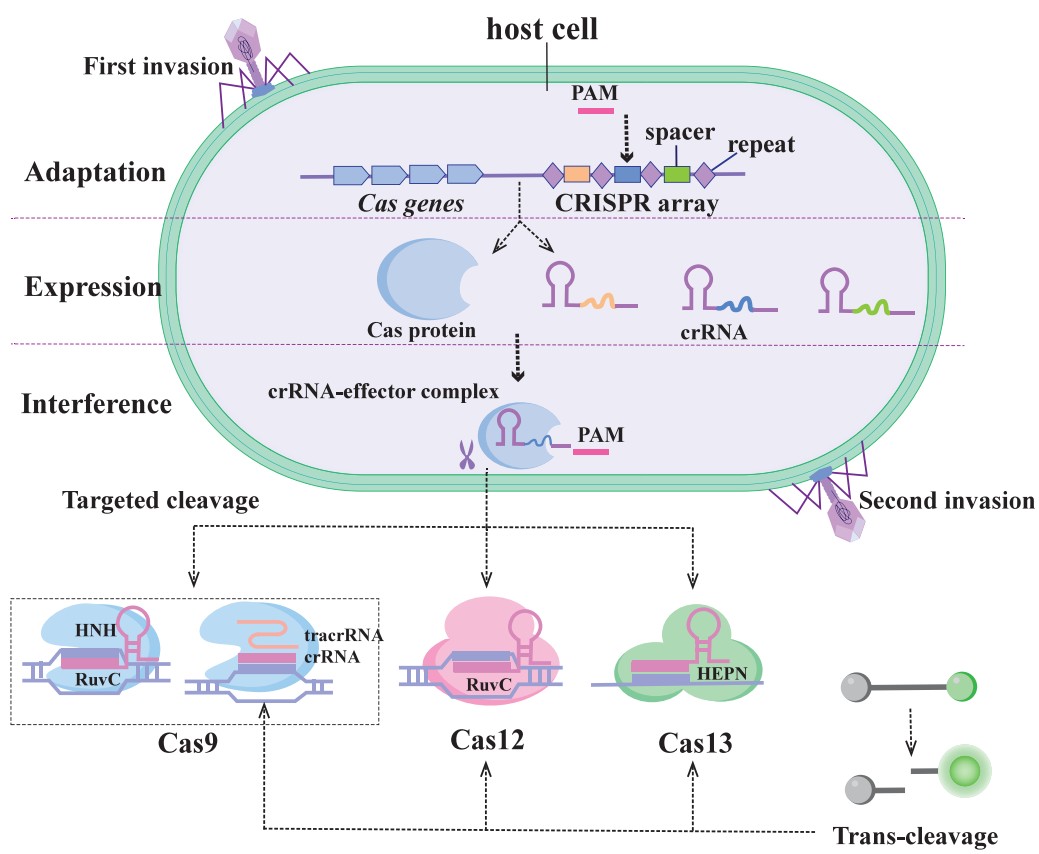

**Figure 1** **Immunocleavage mechanism of the CRISPR/C as system.**

## CLEAVAGE CHARACTERISTICS OF THE CRISPR/CAS SYSTEM

The CRISPR/Cas system was first identified in *E. coli* as an adaptive immune system in prokaryotes, used to protect against the threat of foreign genetic material (Fig. 1) (*Marraffini, 2015*). The system uses the guide RNA (gRNA) sequence encoded in the CRISPR locus to guide the Cas protein to recognize and cleave specific foreign nucleic acid sequences. CRISPR/Cas systems based on the specific recognition and targeted cleavage of foreign genes have been rapidly developed and applied in the fields of gene editing and molecular diagnostics (*Perez-Lopez & Mir, 2021*).

The CRISPR/Cas system is divided into two categories of six types: Class 1 (types I, III, and IV) and Class 2 (types II, V, and VI) (*Koonin, Makarova & Zhang, 2017*). The Class 1 system features multiprotein effector complexes, whereas the Class 2 system comprises multidomain single-effector proteins responsible for pre-crRNA processing, crRNA binding, and nucleic acid cleavage (*Makarova et al., 2020*). Among them, the Class 2 system has more biotechnology applications and nucleic acid detection capabilities and is also easy to use (*Fapohunda et al., 2022*). These include the enzymes Cas9 (type II), Cas12 (type V), and Cas13 (type VI), which have incidental activity and constitute the backbone

of many CRISPR-based diagnostics. Therefore, the second class of CRISPR systems is described in this paper.

### Type II (Cas9)

CRISPR/Cas9 is a binary complex consisting of the effector protein Cas9 and single-guide RNA (sgRNA), which may target and cleave dsDNA in cis. The sgRNA consists of complementary binding of CRISPR RNA (crRNA) and transactivating CRISPR RNA (tracrRNA), which can guide Cas9 to target the DNA sequence next to the protospacer adjacent motif (PAM) through complementary base pairing (*Garneau et al., 2010*). Its PAM is crucial for both the acquisition of a new spacer and the subsequent identification and silencing of the target DNA. Structural analysis revealed that the effector protein Cas9 is dependent on two endonuclease domains, namely, the HNH domain located in the middle region of the protein and the RuvC domain located at the amino terminus (*Jinek et al., 2014*), which are responsible for the cleavage of the target DNA strand (TS) and nontarget strand (NTS), respectively (*Jiang et al., 2016*). In recent years, studies have reported that Cas9 exhibits only the cis-cleavage ability on dsDNA but cannot perform *trans*-cleavage, which is the ability to cleave any ssDNA or ssRNA within the reaction system. However, a recent study revealed that Cas9 *trans*-cleavage can be activated when it is codirected by crRNA and tracrRNA, contradicting the conventional knowledge that Cas9 does not possess *trans*-cleavage activity (*Chen et al., 2024a*). This discovery further expands the range of applications of Cas9.

Although the CRISPR/Cas9 system has yielded numerous impressive discoveries in gene editing and other fields, one of the most significant barriers to its widespread adoption in the future is the safety risk posed by its off-target effects.

### Type V (Cas12 and Cas14)

The Cas protein family of the Type V system includes Cas12a∼Cas12n, which have endonuclease activity but lacks the HNH domain. In addition, the Type V Cas family can cleave any ssDNA sequence in the reaction system, a process known as *trans*-cleavage, which is activated after the target sequence is identified and cleaved. Regardless of whether the target is ssDNA or dsDNA, *trans*-cleavage of Cas12 can be triggered. When ssDNA is targeted, the cutting speed is faster without PAM restriction (*Li et al., 2020*); when dsDNA is used as a target, the target dsDNA needs to contain a PAM (*Yuan et al., 2022*).

Cas12a (Cpf1) is a member of the V-A-type CRISPR/Cas system, the first member of the Cas12 family to be identified, and is the most extensively studied. Only a single crRNA is required to guide the identification of target DNA, and tracrRNA is not required (*Zetsche et al., 2015*). More compact Cas12 proteins, such as Cas12b and Cas12f (also known as Cas14a), require both crRNA and tracrRNA (*Strecker et al., 2019*; *Yang et al., 2021*). In addition to highly specific dsDNA cleavage (*cis*-cleavage), Cas12a also shows *trans*-cleavage activity (*Zetsche et al., 2015*). When crRNA attaches to complementary target DNA, Cas12a cleaves any ssDNA molecule into single/dinucleotides (*Chen et al., 2018*). In addition, recent studies have reported that Cas12a can detect ssRNA by recognizing fragmented RNA/DNA targets (*Moon & Liu, 2023*; *Rananaware et al., 2023*). Cas12b (C2c1) belongs to

the V-B CRISPR/Cas system, and its most notable feature is that it is more thermally stable than other family members.

Cas12f (Cas14a), a V-F type, is a newly discovered DNA-targeting CRISPR effector protein characterized by a compact structure and minimal size (*Harrington et al., 2018*). Compared with other Cas12 enzymes, Cas14 is less tolerant of nucleotide mismatches between crRNA and target templates, enabling high-fidelity single-nucleotide polymorphism (SNP) genotyping (*Khan et al., 2019*). Despite its modest size, Cas14a can accomplish specific single-stranded DNA (ssDNA) cleavage without requiring a PAM and can then stimulate the cleavage of nonspecific ssDNA molecules (*Harrington et al., 2018*). Currently, Cas14a is integrated into the DETECTR platform to generate a new ssDNA detection system called Cas14a-DETECTR (*Harrington et al., 2018*). The detection system enables the detection of ssDNA pathogens by Cas14a.

Unlike Cas9, Cas12 can cleave both double-stranded DNA (dsDNA) and single-stranded DNA (ssDNA), and its *trans*-cleavage activity enables the CRISPR/Cas system to pair with fluorescent reporters, allowing for the visualization of the results of pathogen detection. In addition, the off-target effects of Cas12 improved, but the cleavage efficiency decreased. Both Cas12 and Cas9 have advantages, and it is necessary to choose the most suitable protein according to their respective characteristics on different occasions (*Kim et al., 2016*).

### Type VI (Cas13)

Cas13 is a family of RNA-targeted ribonucleases that exhibit target RNA-activated substrate RNase activity (*Abudayyeh et al., 2017*). Cas13 has two distinct ribonuclease activities. One RNase activity is responsible for precrRNA processing to help form a mature type VI interference complex; two higher eukaryotic and prokaryotic nucleotide-binding domains (HEPNs) provide further RNase activity by efficiently cleaving target RNA (*Anantharaman et al., 2013*). PAM sequences do not limit the specific recognition of RNA by Cas13-crRNAs but usually rely on the presence of a protospacer flanking sequence (PFS) at the 3ʹend of the target ssRNA. Cas13a also has *trans*-cleavage activity and can cleave arbitrary nontargeted RNA (*East-Seletsky et al., 2016*). Compared to the other two types of Cas proteins, the limiting sequence PFS of Cas13 is composed of A, U, or C, which increases the fault tolerance of Cas13. Cas13 can tolerate single-base mismatches between the crRNA and target sequence without reducing cleavage effectiveness (*Zhong et al., 2017*). However, a recent study revealed that DNA can serve as a direct target for CRISPR-Cas13a to activate the *trans*-RNase activity of Cas13a without being restricted by PFS and PAM sequences (*Wu et al., 2024b*). This finding overcomes the limitation that Cas13 can recognize only RNA and holds significant value in future CRISPR diagnosis. The functions and characteristics of the three types are shown in Table 1.

## THE PROSPECTS OF CRISPR TECHNOLOGY IN PATHOGEN DETECTION

Due to the complexity and diversity of pathogen infections, the emergence of new pathogens, and the increasingly severe situation of drug resistance caused by microbial

**Table 1  Comparison of Type II, Type V, and Type VI systems.**

| Type | II | | V | | VI |
|---|---|---|---|---|---|
| **Effector protein** | **Cas9** | **Cas12a** | **Cas12b** | **Cas14a** | **Cas13a** |
| Nuclease domains | HNH, RuvC | RuvC | RuvC | RuvC | HEPN |
| TracrRNA | Yes | No | Yes | Yes | No |
| Target | dsDNA, ssDNA, ssRNA | dsDNA, ssDNA, RNA/DNA | dsDNA, ssDNA | ssDNA | dsDNA, ssDNA, ssRNA |
| Trans-cleavage substrates | ssDNA, ssRNA | ssDNA | ssDNA | ssDNA | ssRNA |
| PAM sites | NGG | TTTN | TTTN | – | PFS |

gene mutations, the effective prevention and control of infectious diseases are significantly hindered (*Fisher, 2021*). Currently, clinical mutation detection methods include sequencing, quantitative PCR (qPCR), and digital droplet PCR (ddPCR) (*Ou et al., 2023*; *Winters et al., 2024*). However, sequencing is associated with high costs and significant time investment, and its sensitivity is contingent upon sequencing depth. Additionally, probe-based methods exhibit limited sensitivity, particularly for analyzing single nucleotide variations. Due to its highly accurate target identification capabilities and programmability, the CRISPR/Cas system has revolutionized the field of gene editing and paved the way for new directions *in vitro* diagnostics. CRISPR gene editing relies on Cas9 or Cas12a for site-specific recognition and cleavage of target DNA, resulting in double-stranded DNA breaks (DSBs). DNA repair is performed *in vivo* by either error-prone nonhomologous end joining (NHEJ) or error-free homology-directed repair (HDR). NHEJ often leads to random DNA insertion deletions at the cleavage site, whereas HDR performs precise sequence insertions or gene replacements by adding donor DNA templates with sequence homology at the predicted DSB sites (*Char et al., 2017*). In CRISPR diagnosis, the design of optimized and more mismatched-tolerant guide RNA and Cas proteins, pathogen mutation detection with single-nucleotide resolution can be achieved.

For example, *Chavez et al. (2018)* endowed Cas9 with the ability to distinguish single nucleotide polymorphisms (SNPs) by designing a tuning guide RNA (tgRNA) containing a single nucleotide mismatch, which was successfully used to prevent point mutations in *E. coli* resistance genes. Subsequently, *Teng et al. (2019)* used this optimized tgRNA to develop a Cas12b-based precise DNA detection platform (CDetection) with single-base resolution, enabling the differentiation of single-base differences and further improving the specificity of CRISPR mutation detection. Multiple mutations typically characterize pathogen mutations; therefore, by combining a multiplexing strategy, CRISPR enables fast and efficient analysis of pathogen variants. For example, the CARMEN (combinatorial arrayed reactions for multiplexed evaluation of nucleic acids) platform (*Ackerman et al., 2020*) combines Cas13-based nucleic acid detection with microtiter arrays to empower CRISPR high-throughput assays that are able to simultaneously distinguish 169 viruses relevant to human health and enable comprehensive subtyping of influenza A strains as well as the identification of dozens of HIV-resistant mutants. However, a recent study

reported a method for detecting multiple mutations in single-crRNA using the engineered AsCas12a protein, which can identify multiple insertions/deletions at the same genetic locus *via* a single target crRNA (*Liu et al., 2024*). This method further advances the immediate detection of pathogen mutations and will play an important role in future mutant screening. Additionally, the current CRISPR gene editing technology has made significant progress in the field of pathogenic microbial genome research. The precise identification of virulence- and resistance-related genes generated by mutations *via* CRISPR provides not only a basis for drug therapeutic targets but also new targets for mutation detection, pathogen mutation detection (*Bosch et al., 2021*; *Lai et al., 2021*).

Owing to the unique advantages of the CRISPR/Cas system, which is highly accurate and easy to use, CRISPR diagnostic tools have been gradually developed and applied for pathogen mutation detection and drug resistance monitoring and will play a key role in future global pathogen prevention and control. Although CRISPR diagnosis has undergone significant evolutionary expansion, additional efforts are still needed to accelerate its clinical application. To facilitate this application, several optimization strategies have been implemented, including simplified sample preparation, streamlined assay processes, and portable readout strategies, to avoid the various shortcomings present in routine CRISPR assays.

### The simplified strategies for CRISPR pathogen nucleic acid detection

Although early CRISPR detection platforms (SHERLOCK and DETECTR) have achieved relatively mature detection results in terms of sensitivity and specificity, further improvements can still be made for clinical applications. For example, additional sample processing, amplification, and liquid transfer steps are required, which undoubtedly increase the complexity of detection, the risk of cross-contamination, and aerosol generation. In addition, signal readouts based on complex and expensive fluorescence detection systems limit their application in field detection (*Gootenberg et al., 2018*).Therefore, various optimized, simplified strategies for CRISPR-based diagnostics have been proposed, shedding new light on pathogen detection and paving the way for the application of point-of-care testing (POCT) in the field of rapid pathogen detection (Table 2). The advantages and disadvantages of various strategies are summarized in Table S1.

### Nucleic acid extraction-free detection strategy

The nucleic acid extraction step is not only the speed-limiting part of POCT but also one of the key problems associated with the CRISPR/Cas assay. Solid-phase/liquid-phase extraction has high extraction efficiency but requires multiple transfer steps and additional instrumentation and technical expertise, which significantly impact the overall detection time, cost and accuracy of the results (*Dineva, MahiLum-Tapay & Lee, 2007*). Therefore, several methods of nucleic acid-free extraction are summarized below to promote the excellent simplicity of CRISPR detection technology.

**Table 2  The application of the simplified CRISPR strategy for pathogen detection.**

| Method | Platform name | Cas protein | Pathogens | Combined amplification technology | Assaying time | Sensitivity | References |
|---|---|---|---|---|---|---|---|
| | | | Extraction-free detection strategy | | | | |
| | SHERLOCK | Cas13a | DENV | RPA | <2 h | 1 copy/µL | *Myhrvold et al. (2018)* |
| | SHINE | Cas13a | SARS-CoV-2 | RPA | 50 min | / | *Arizti-Sanz et al. (2022)* |
| Heat/chemical cracking | S- PREP | Cas13a | *Plasmodium* | RT-RPA | 1 h | <two parasites/ microliter blood | *Lee et al. (2020)* |
| | SHINEv.2 | Cas13a | SARS-CoV-2 | RPA | <90 min | 200 copies/µL | *Arizti-Sanz et al. (2022)* |
| Magnetic bead purification | DNA-FISH | dCas9 | MRSA | \ | <30 min | <10 CFU/ml | *Guk et al. (2017)* |
| | STOPCovid | Cas12b | SARS-CoV-2 | RT-LAMP | <1 h | 100 copies | *Joung et al. (2020)* |
| Others | miSHERLOCK | Cas13a | SARS-CoV-2 | RPA | 1 h | \ | *De Puig et al. (2021)* |
| | SPEEDi-CRISPR | Cas12a | HPV-18 | \ | \ | \ | *Tian et al. (2021)* |
| | | | Amplification-free detection strategy | | | | |
| | E-CRISPR | Cas12a | HPV-16 and PB-19 | \ | 30 min | 1 pM | *Dai et al. (2019)* |
| | E-DNA | Cas9 or Cas12a | PB-19 | \ | \ | 1 fM | *Xu et al. (2020)* |
| Signal conversion strategy | E-Si-CRISPR | Cas12a | MRSA | \ | 1.5 h | 3.5 fM | *Suea-Ngam, Howes & De Mello (2021)* |
| | CRISPR/ Cas13a-gFET | Cas13a | SARS-CoV-2 and RSV | \ | <30 min | 1 aM | *Li et al. (2022a)* |
| | \ | Cas12a | HPV | \ | \ | 10 fM | *Fu et al. (2021)* |
| | CRISPR/ dCas9-SERS | dCas9 | HPV | \ | \ | ng | *Su et al. (2023)* |
| Signal enhancement strategy | CRISPR/ Cas12a-SERS | Cas12a | Virus DNA | \ | <20 min | 1 aM | *Choi et al. (2021b)* |
| | CRISPR/ Cas12a-SERS | Cas12a | HBV | \ | <50 min | 0.1 pM | *Du et al. (2023)* |
| | \ | Cas13a | Bacteria 16S rRNA and Virus RNA | \ | \ | 1 aM | *Tian et al. (2021)* |
| | \ | Cas12a | ASFV | \ | \ | 17.5 copies/µL | *Yue et al. (2021)* |
| | SATORI | Cas13a | SARS-CoV-2 | \ | 5 min | 5 fM | *Shinoda et al. (2021)* |
| Target/signal enrichment strategy | opn-SATORI | Cas13a | SARS-CoV-2 | \ | 9 min | <6.5 aM | *Shinoda et al. (2022* |
| | \ | Cas13a | H1N1 and SARS-CoV-2 | \ | <50 min | 2 aM | *Wang et al. (2023)* |
| | Cas-DSM | Cas12a or Cas13a | SARS-CoV-2 and *S. aureus* | \ | \ | 1 aM | *Chen et al. (2023a)* |
| Cascade signal amplification strategy | FIND-IT | Cas13a and Csm6 | SARS-CoV-2 | \ | 20 min | 30 copies/µL | *Liu et al. (2021)* |
| | CONAN | Cas12a | HBV | \ | \ | 5 aM | *Shi et al. (2021)* |
| | | | Integrated response strategy | | | | |
| Traditional one-pot detection | HOLMESv2 | Cas12b | JEV | RT-LAMP | 1 h | \ | *Li et al. (2019)* |
| | CRISPR-SPADE | Cas12b | SARS-CoV-2 | RT-LAMP | <30 min | \ | *Nguyen et al. (2022)* |

**Table 2** (*continued*)

| Method | Platform name | Cas protein | Pathogens | Combined amplification | Assaying time | Sensitivity | References |
|---|---|---|---|---|---|---|---|
| | AIOD-CRISPR | Cas12a | SARS-CoV-2 | RPA | 40 min | 5 copies | *Ding et al. (2020)* |
| Optimized one-pot detection | sPAMC | Cas12a | Human cytomegalovirus and SARS-CoV-2 | RPA | 20 min | \ | *Lu et al. (2022)* |
| | \ | Cas12a | SARS-CoV-2 | RT-RPA | \ | 100 copies | *Hu et al. (2022)* |
| | \ | Cas12a | HPV-16 | RPA | \ | 0.2 copies/μL | *Lesinski et al. (2024)* |
| Microfluidic integrated detection | \ | Cas12a | *S. aureus* | RPA | 55 min | 32 CFU/mL | *Lu et al. (2023)* |
| | \ | Cas12a | SARS-CoV-2 | RT-RPA | \ | 100 copies | *Li et al. (2022c)* |
| | \ | Cas13a | virus | RPA | 45 min | 1 copy | *Zhang et al. (2024a)* |
| | | | Simple output strategy | | | | |
| Ultrasensitive lateral chromatography system | SHERLOCKv2 | Cas13a, Cas12a, Csm6 | ZIKV and DENV | RPA | <2 h | 2 aM | *Gootenberg et al. (2018)* |
| | CRA-LFB | Cas12a | *S. aureus* | RAA | 70 min | 75 aM | *Zhou et al. (2022)* |
| | CQ-LFA | Cas12a | VZV | RAA | <1 h | 5 copies | *Zhong et al. (2023)* |
| Smartphone readout system | \ | as13a | SARS-CoV-2 | \ | 30 min | 170 aM | *Fozouni et al. (2021)* |
| | \ | Cas12a | *Salmonella* | \ | \ | 1 CFU/mL | *Yin et al. (2021)* |
| | \ | Cas9 | SARS-CoV-2 | LAMP | <1 h | 1 aM | *Song et al. (2022)* |
| Integrated small readout system | \ | Cas12a | ASFV | \ | <2 h | 1 pM | *He et al. (2020)* |
| | \ | Cas12a | *Alternaria* | \ | \ | 1.5 pM | *Liu et al. (2022)* |
| | CRISPR-MCR | Cas12a | HIV | RT-RPA | \ | 200 copies | *Li et al. (2023b)* |

### Heating/chemical pyrolysis methods

In 2018, *Gootenberg et al.*'s (*2018*) team proposed a scheme called heating unextracted samples to eliminate nucleases technology (HUDSON), which utilizes heating and chemical reduction to lyse virus particles and inactivate high concentrations of RNase, thereby eliminating the nucleic acid extraction step in the detection process (Fig. 2A) (*Myhrvold et al., 2018*). HUDSON is applicable to a variety of clinical samples (including urine, whole blood, plasma, serum, and saliva) and can destroy viral particles and inactivate RNase within 10 min without dilution or purification and without affecting subsequent amplification or detection. The protocol has been widely applied to SHERLOCK, DETECTR, Streamline Highlights of Infections to Navigate Epidemics (SHINE), and other major detection platforms (Table 2) (*Aquino-Jarquin, 2019*; *Arizti-Sanz et al., 2020*; *Myhrvold et al., 2018*). However, HUDSON cannot achieve good detection results when facing relatively high concentrations of nuclease. Therefore, the SHERLOCK Rapid Parasite Extraction Protocol (S-PREP) was proposed for high-specificity parasite detection without the need for nucleic acid extraction (*Lee et al., 2020*). In the S-PREP scheme, Chelex-100 (with paired iminodiacetate ions) was first used as a chelating agent that can bind polyvalent metal ions. Because a nuclease requires metal ions as cofactors, the chelating agent can inhibit its activity. In this case, the enzyme can then be heated at 95 °C for 10 min to inactivate the nuclease. It can inactivate a high level of nuclease without nucleic acid extraction, and its detection specificity for Plasmodium species reaches 100%. In addition, to eliminate the equipment needed for sample heating, the sample was treated with FastAmp lysis reagent supplemented with 5% RNase inhibitor in SHINEv.2, and the RNase activity was reduced

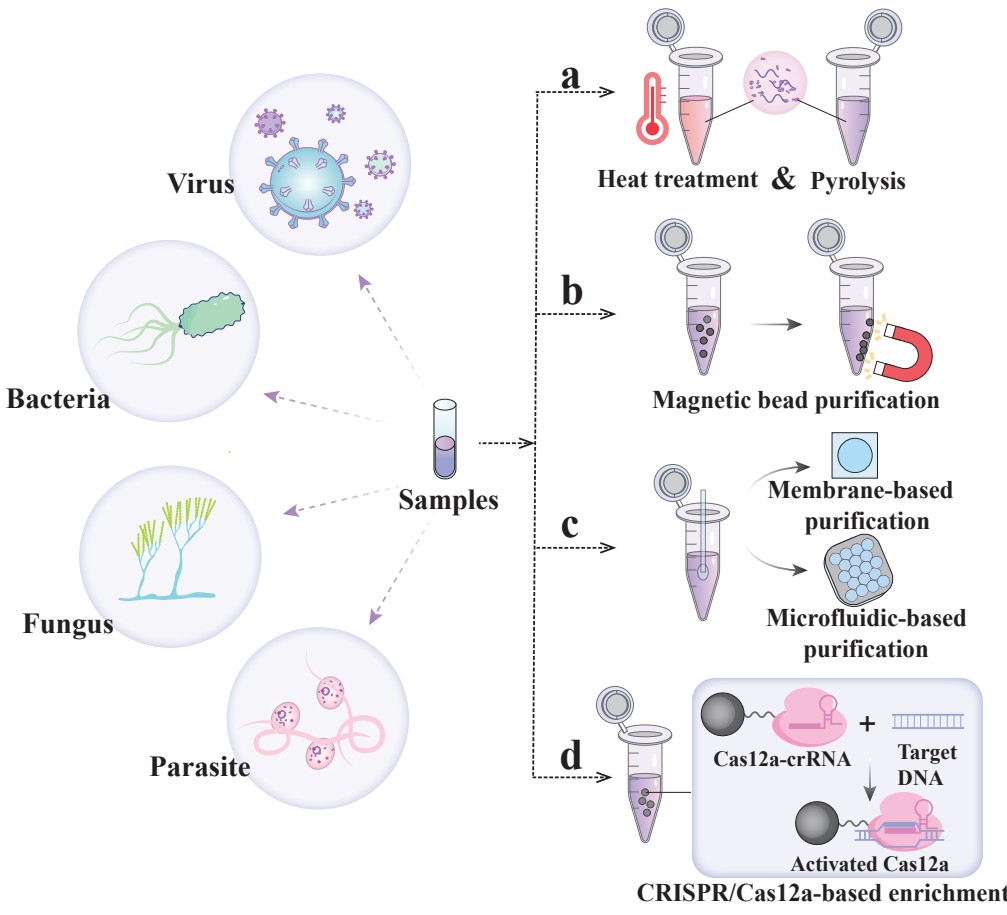

**Figure 2  Nucleic acid extraction-free.**

to 85% at room temperature (*Arizti-Sanz et al., 2022*). Compared with SHINE, SHINEv.2 eliminates the need for heating steps and reagent refrigeration, enabling equipment-free SARS-CoV-2 detection with a sensitivity of 200 copies/μL.

Although the heating/chemical pyrolysis method is compatible and has been widely verified in various pathogen tests, the lack of nucleic acid purification and concentration steps may compromise the ability of CRISPR to detect microtargets in complex matrix samples.

### *Magnetic bead purification method*

To further enhance the sensitivity of extraction-free CRISPR/Cas detection for trace pathogen nucleic acids, it is necessary to employ simple and efficient methods for separating and purifying cell lysates from complex matrices. Nanoscale micromagnetic beads possess unique surface characteristics and can be rapidly immobilized in the target region to facilitate the separation and purification of nucleic acids released into sample lysates (Fig. 2B) (*Chen et al., 2021*). For example, in CRISPR-mediated DNA-FISH, Ni-NTA magnetic beads were used to pull out the cell lysate sample directly, eliminating the need

for gene purification and the subsequent nucleic acid extraction step (*Guk et al., 2017*). In addition, in the SHERLOCK One-Pot testing (STOP), COVID-19 uses magnetic bead adsorption to concentrate the SARS-CoV-2 RNA genome into a STOPCovid reaction mixture, thus eliminating the ethanol washing and RNA elution steps and shortening the sample extraction time to 15 min with the least manual operation time (Table 2) (*Joung et al., 2020*).

Magnetic bead purification can quickly concentrate and purify the target from large-volume samples without tedious operation steps, thereby improving the sensitivity of detection and increasing its applicability, which further promotes the field deployment of CRISPR/Cas detection. However, magnetic bead purification carries the risk of bead residue and is dependent on cost and equipment, with significant consumable costs in high-throughput applications.

### Others

In addition to nucleic acid purificatio using micromagnetic beads, nucleic acid isolation membranes formed by nanoscale or microscale fibers can also be employed for efficient nucleic acid purification and concentration from many samples at a lower cost than the magnetic bead purification method (Fig. 2C) (*Rodriguez et al., 2015*). The principle involves rapid and efficient separation of nucleic acids from lysed samples in high-salt solutions *via* electrostatic interactions and hydrogen bonding. The nucleic acids adsorbed on these membranes bind stably *via* simple washing, allowing for the removal of pollutants, and can be amplified *in situ* without elution. For example, with the minimum-instrument SHERLOCK (miSHERLOCK), SARS-CoV-2 RNA was isolated from two mL of saliva using gravity and capillary action through a four mm polyether sulfone (PES) membrane (Table 2) (*De Puig et al., 2021*). The simplicity of this design enables instrument-free, intuitive liquid handling, achieving significant sample concentrations that enhance the overall signal by a factor of 2 to 20. Further studies have reported that microfluidic technology enables the rapid purification and concentration of nucleic acids in small-volume samples (Fig. 2C) (*Obino et al., 2021*). This technology can integrate sample preparation, amplification, the CRISPR reaction, and signal readout steps into a microfluidic platform, simplifying detection and potentially improving its analytical and diagnostic performance (*Ramachandran et al., 2020*).

In addition, the solid-phase extraction and enhanced detection assay, integrated with CRISPR-Cas12a (SPEEDi-CRISPR), utilizes the sequence recognition of CRISPR/Cas12a for nucleic acid extraction for the first time, integrating CRISPR/Cas enrichment and detection capabilities into a single platform (Fig. 2D) (*Tian, Yan & Zeng, 2024*). Cas12a-crRNA RNPs are first bound to the surface of magnetic beads, and the target DNA in the sample is captured on the beads through sequence recognition by the RNP, activating the *trans*-cleavage activity of Cas12a and generating a signal through the cleavage of the ssDNA reporter. The platform is capable of detecting human papillomavirus 18 (HPV-18) at concentrations as low as 2.3 fM in 100 min and 4.7 fM in 60 min. The simplicity and compatibility of its equipment facilitate POC testing.

## Nucleic acid amplification-free detection strategy

Nucleic acid amplification-free strategies can not only greatly simplify operation procedures and reduce the use of expensive equipment but also avoid residual aerosol pollution and amplification bias, improving the accuracy of detection. Moreover, it can directly quantify the unamplified target sequence, which optimizes the detection performance of the CRISPR/Cas assay. However, due to the low target concentration of early pathogen infection, a single amplification-free CRISPR/Cas assay may yield extremely weak readouts, making it challenging to achieve clinical detection (*Arnaout et al., 2021*; *European Association for the Study of the Liver, 2018*). Therefore, to improve detection sensitivity, it is usually necessary to combine various advanced biotechnologies for signal conversion, signal enhancement or target/signal enrichment or to adopt a cascade signal amplification strategy to achieve amplification-free detection.

### *Signal conversion strategy*

Electrochemical technology is an ultrasensitive detection technology that can convert the extremely weak signal produced by CRISPR/Cas *trans*-cleavage into a detectable signal, enabling low-cost, rapid and simple ultrasensitive detection. *Dai et al. (2019)* combined an electrochemical biosensor with the CRISPR/Cas system for the first time to develop a CRISPR/Cas12a-based electrochemical biosensor (called E-CRISPR). Methylene blue (MB)-modified ssDNA was paired on the electrode surface. When the target nucleic acid is present, complementary Cas12a-crRNA binds to the target DNA and activates its *trans*-cleavage activity. The ssDNA-MB was cleaved, and the MB was released, resulting in electron transfer that allowed the target to be detected directly by monitoring the change in electrochemical current. The limit of detection (LOD) of human papillomavirus 16 (HPV-16) and parvovirus B19 (PB-19) by this platform reach picolar levels. The same team then proposed a CRISPR/Cas system-enhanced electrochemical DNA sensor (E-DNA), which can induce a conformational change in the surface signal probe (containing the electrochemical tag MB) upon binding to the target, resulting in a change in the electron transfer rate of the electrochemical tag (*Xu et al., 2020*). The platform utilizes hairpin DNA as a signal probe, which causes the MB to move closer to the surface of the gold electrode, thereby increasing the current response. *Cis*-cleavage mediated by Cas9 or Cas12a resulted in the separation of MB from the probe after recognition of the PAM and target sequence, leading to a change in the electron transfer rate of MB. Consequently, the detection sensitivity of PB-19 reached the fM level. Another study combined silver metallization with Cas12a to propose a new silver-enhanced E-CRISPR biosensor (E-Si-CRISPR) (*Suea-Ngam, Howes & De Mello, 2021*). The ssDNA fixed on the electrode is cleaved by the Cas12a:crRNA:target DNA ternary complex, followed by silver metallization, and the degree of silver deposition is proportional to the amount of remaining ssDNA. The final electrochemical signal can be read *via* square wave voltammetry (SWV). Using this platform, the sensitivity of the MRSA mecA gene was improved to the fM level.

Graphene field-effect transistors (gFETs), as sensitive and advanced electronic devices, can achieve hypersensitive signal conversion and amplification. A field-effect transistor (FET) is a three-terminal electronic device with a gate that can regulate the source–drain

current through the semiconductor channel by applying an external electric field to the gate (*Liang et al., 2020*). Graphene is a low-cost zero-gap material with extremely high electron carrier mobility and has been used for high-current-gain FETs. *Bruch, Urban & Dincer (2019)* proposed a biosensor based on CRISPR/dCas9-gFETs. The detection principle involves fixing the target-specific dCas9-crRNA on the surface of gFETs and regulating the electrical characteristics of gFETs through specific binding between the target and the dCas9-crRNA complex for sensor signal transduction. The platform can produce an electrical output with a LOD of 1.7 fM within 15 min. Later, *Li et al. (2022a)* developed a biosensor based on CRISPR/Cas13a-gFET, which fixes a negatively charged RNA reporter (PolyU20) on gFET. When the target is present, the *trans*-cleavage activity of Cas13a is activated, the ssRNA reporter is shed, and a decrease in the number of electron carriers in the graphene channel is observed. Compared with that of CRISPR/dCas9-gFET, the multi-turnover *trans*-cleavage of Cas13a leads to high-speed signal generation, which can detect SARS-CoV-2 and respiratory syncytial virus (RSV) genomes as quickly as 1 aM.

### Signal enhancement strategy

The key factor in ultrasensitive detection is to maximize the detection signal intensity. Gold nanoparticles, as highly sensitive nanomaterials, can achieve amplification-free detection by enhancing the signal readout. *Choi et al. (2021a)* utilized ssDNA-functionalized AuNPs and CRISPR/Cas12a-mediated *trans*-cleavage to achieve target amplification-free detection, employing the principle of metal-enhanced fluorescence. The target DNA sequence activates the CRISPR/Cas12a complex, which subsequently degrades the ssDNA between the AuNPs and the fluorophores, resulting in a colour change from purple to reddish purple. The system can achieve high-sensitivity detection within 30 min. In addition, *Fu et al. (2021)* designed a spherical nucleic acid (SNA) reporter using gold nanoparticles and developed a stable and sensitive CRISPR assay. An SNA reporter was formed by immobilizing fluorescently labelled ssDNA on the gold nanoparticle (AuNP) core. Due to the high quenching efficiency of AuNPs for fluorescent chromophores, the platform enables high signal-to-noise ratio DNA detection at the femtomolar (fM) level. In addition, nanoenzymes are nanomaterials with enzyme-like catalytic activity that play a crucial role in enhancing the sensitivity of CRISPR-based detection, enabling both visual and rapid CRISPR detection (*Zhang et al., 2024b*). Labeling recognition molecules (such as antibodies and DNA) with nanoenzymes instead of natural enzymes can enhance signal readout, resulting in highly sensitive colorimetric or ratio fluorescence sensors (*Chi, Wang & Gu, 2023*; *Wu et al., 2024a*).

Surface-enhanced Raman scattering (SERS), a signal enhancement technique, can manipulate the surface plasmon effect on a metal surface and further enhance the Raman vibration signal of the target, allowing for the detection of low-concentration biomolecules with extremely high sensitivity (*Dai et al., 2021*). *Su et al. (2023)* combined the CRISPR/dCas9 system with SERS detection and enzyme-catalyzed reactions to develop a CRISPR-SERS assay. The CRISPR/dCas9/sgRNA complex immobilized on magnetic beads facilitates DNA enrichment and separation. Horseradish peroxidase (HRP) was introduced into biotinized target DNA to catalyze the substrate TMB (3,3′,5,5′-tetramethylbenzidine)

to produce a blue–green oxidation product (oxTMB), which can provide colorimetry and SERS signal output, and the SERS spectrum of oxTMB was measured by gold nanostars with a silica shell to achieve double amplification. This platform can rapidly detect the ng level of HPV-DNA without the need for preamplification.

In addition, the sensitivity of SERS-based detection can be further enhanced by utilizing the surface plasmon effect of precious gold nanoparticles. *Choi et al. (2021b)* developed an amplification-free CRISPR/Cas12a-SERS sensitive detection platform based on Raman probe-functionalized gold nanoparticles (RauNPs). In this platform, ssDNA-RauNP probes are immobilized on graphene oxide (GO)/triangular gold nanoflower (TANF) arrays to enhance the SERS signal through GO-TANF arrays. In the presence of target viral DNA, the CRISPR/Cas12a complex was activated to cleave ssDNA between RauNPs and GO-TANFs. The platform can detect hepatitis B virus (HBV), HPV-16 and HPV-18 with a sensitivity of 1 aM within 20 min. *Du et al. (2023)* combined gold nanoparticles (AuNPs) with magnetic spheres as Raman probe molecules and proposed a CRISPR/Cas12a-based surface-enhanced Raman spectroscopy assay. The SERS signal was enhanced by the AuNPs modified with the Raman reporter molecule 4-ATP. AuNPs were attached to the magnetic spheres *via* ssDNA to achieve the maximum SERS effect and a differential Raman spectral signal. The target gene could activate the Cas12a/crRNA complex to cleave ssDNA attached to the AuNPs and magnetic spheres. Therefore, the system can detect HBV DNA rapidly and sensitively within 50 min.

### Target/signal enrichment strategy

Using ultralocal or closed microvolume reactors (such as droplet microfluidics, microchamber arrays, and micromagnetic beads) to highly aggregate local, low-concentration targets or signals can achieve single-molecule sensitivity detection. For example, *Tian et al. (2021)* proposed a CRISPR/Cas13a assay based on the droplet microfluidic restriction effect for single-molecule RNA diagnosis without amplification or reverse transcription. The platform utilizes droplet microfluidics to confine the target RNA-triggered Cas13a catalytic system in a cell-sized reactor, thereby simultaneously increasing the local concentrations of both the target and the reporter. Compared to batch Cas13a detection, the sensitivity is improved by more than 10,000 times, enabling absolute digital quantification of single-molecule RNA. The same research group further proposed a double-crRNA droplet-based CRISPR/Cas12a assay for amplification-free and absolute quantification of DNA at the single-molecule level (*Yue et al., 2021*). The platform employed a dual crRNA strategy to enhance reaction efficiency and performed Cas12a reactions in pL water-oil droplets, which can directly detect 17.5 copies/µL of African swine fever virus (ASFV).

However, the droplet has unstable performance, which may affect detection. Compared with the droplet method, the microchamber array has a more uniform reaction volume, and its fL volume results in a better concentration effect (*Cohen et al., 2020*). *Shinoda et al. (2021)* combined microchamber array technology with CRISPR/Cas13a to develop a CRISPR-based amplification-free digital RNA detection platform (SATORI). Similarly, a microchamber array can increase the local molecular concentration and accelerate

detection. The device contains more than $1 \times 10^6$ through-holes and can observe a large number of chemical reactions in parallel at the single-molecule level, enabling the detection of SARS-CoV-2 in as little as 5 min with a sensitivity of 10 fM. By combining multiple crRNAs, the sensitivity can be increased to 5 fM. However, some manual processing procedures may introduce human errors, which may lead to reduced detection accuracy of SATORI. Therefore, a fully automated platform for SATORI (opn-SATORI) is proposed to avoid complex solution exchange steps (*Shinoda et al., 2022*). When the optimal Cas13a enzyme and magnetic bead technology are utilized, the limit of detection (LoD) of opn-SATORI for SARS-CoV-2 genomic RNA is less than 6.5 aM, with a sensitivity comparable to that of RT-qPCR. Recently, *Wang et al. (2023)* combined magnetic bead capture with a microchamber array to develop a CRISPR/Cas13a-based single-molecule digital detection method for nucleic acids. The target nucleic acid fragments were captured and enriched with magnetic beads. The target-induced Cas13a cleavage reaction was dispersed and confined to a million single-fL micropores, thereby increasing the local signal intensity and enabling single-molecule detection. The LOD of the method for H1N1 virus and SARS-CoV-2 is 2 aM.

In addition, a study reported the use of a CRISPR/Cas-driven single micromotor (Cas-DSM). The micromotor is used as a universal microreactor for CRISPR/Cas reactions and as a unique readout platform to enrich all the generated fluorescence signals in a very small location (*Chen et al., 2023a*). By fixing the fluorescently labeled ssDNA or RNA on the magnetic micromotor, the *trans*-cleavage capability of the CRISPR/Cas system is activated when the target DNA/RNA is present, and the FQ-ssDNA/ssRNA reporter on the surface of the micromotor is cleaved, resulting in the separation of the fluoro group (FAM) from the quenching group (BHQ1), leaving only FAMs on the micromotor surface, thus illuminating the microreactor under laser light. The platform utilizes highly enriched signals to detect nucleic acid targets at the single-molecule level directly, eliminating the need for amplification steps.

### Cascade signal amplification strategy

Although the above methods enable sensitive detection of pathogens without amplification, detection is dependent on other sensitive sensing systems, which require the discovery of biosensors integrated with CRISPR/Cas systems. The CRISPR/Cas cascade signal amplification strategy (including Cas-mediated and crRNA-mediated cascade amplification) can achieve direct detection of pathogens without combining with other sensing systems and has great application prospects for amplification-free CRISPR/Cas detection.

The Cas protein cascade can improve the sensitivity and specificity of detection and has the potential for amplification-free detection. For example, in SHERLOCKv2, the type III effector protein Csm6, in conjunction with Cas13, can increase the rate of RNA detection by 3.5 times compared with Cas13 alone. Additionally, 4-channel multiplexing is achieved by using three Cas13 enzymes and one Cas12a enzyme (*Gootenberg et al., 2018*). Researchers subsequently proposed fast integrated nuclease detection in the tandem platform (FIND-IT), which integrates Cas13a and Csm6 and can directly detect pathogen

RNA quickly (20 min), with a sensitivity of 30 copies/μl (*Liu et al., 2021*). Csm6 is a dimeric RNA endonuclease. When the target RNA activates Cas13a-crRNA, Cas13a cuts the ssRNA substrate, and the degraded substrate acts as an activator of Csm6, inducing the Csm6 *trans*-cleavage RNA reporter probe to achieve signal cascade amplification.

Cascaded signal amplification initiated by crRNA can also enable direct detection of the target sequence. For example, one study developed a method for the direct detection of SARS-CoV-2 *via* the synergy of crRNA, which was designed to contain more than ten crRNAs and could detect SARS-CoV-2 RNA at concentrations as low as 170 aM within 30 minutes (*Fozouni et al., 2021*). However, this assay, which combines multiple crRNAs, is only suitable for detecting long-strand nucleic acids and detecting long-strand nucleic acids and has limited ability to detect short-strand nucleic acids. Another study proposed the use of the autocatalytic nucleic acid positive feedback network of the CRISPR/Cas12a system, namely, the CRISPR/Cas-only amplification network (CONAN), to overcome these difficulties (*Shi et al., 2021*). To achieve target nucleic acid index signal amplification, the platform builds a feedback amplification network by creating a gRNA for target dsDNA (gRNA-T) and a gRNA hybridized with a ssDNA reporter (scgRNA) (Fig. 3A). The self-reporting ability of scgRNA serves to output amplified fluorescent signals and multiple active gRNA molecules in response to each active Cas12a protein with *trans*-cleavage activity. Each target nucleic acid molecule is converted into an exponentially amplified fluorescence signal in CONAN. Thus, ultrasensitive nucleic acid detection is realized with attomolar sensitivity. Moreover, according to a recent study, full-sized crRNA and split crRNA can amplify cascade signals through their competitive response to Cas12a (Fig. 3B) (*Moon & Liu, 2023*). The asymmetric *trans*-cleaving activity of Cas12a is induced by a competitive response between full-sized crRNA and split crRNA. This competitive CRISPR response triggers a conformational reset of the CRISPR enzyme, significantly enhancing the detection sensitivity of CRISPR-Cas12a without requiring additional DNA amplification steps. Furthermore, researchers have demonstrated that Cas12a can be utilized to identify broken RNA/DNA targets, thereby enabling the direct detection of RNA.

The CRISPR/Cas assay with cascaded signal amplification is simple and fast, and it does not require any special instruments or equipment, which is crucial for the realization of POCT and on-site diagnosis. However, its sensitivity is relatively weak, and improving the compatibility of this multienzyme reaction system to achieve simple and sensitive molecular diagnostics remains a challenge.

## Integrated reaction strategy

Most CRISPR/Cas assays (such as SHERLOCK, DETECTR, and HOLMES) require multiple pipetting steps for sample detection, which undoubtedly complicates the operation and simultaneously increases the risk of cross-contamination, preventing their wide application outside of well-controlled laboratories. The integrated reaction strategy partially alleviates the problem and reduces the detection time, thereby addressing the issue of weak feedback (*Wang et al., 2021*).

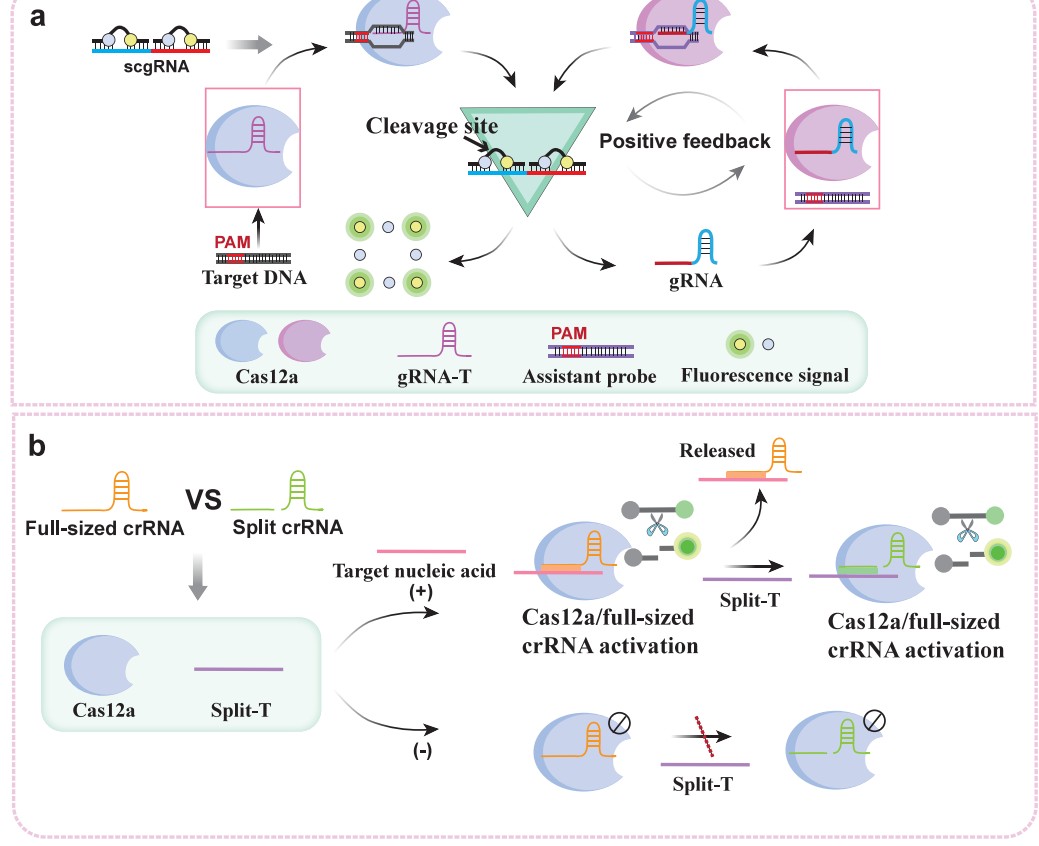

**Figure 3** crRNA-mediated cascade signal amplification strategy: (A) Working principle of the crRNA-mediated CRISPR/Cas12a nucleic acid positive feedback network; (B) working principle of the competitive crRNA-mediated asymmetric CRISPR assay.

### One-pot detection

One-pot detection integrates nucleic acid amplification and the CRISPR/Cas reaction into a single buffer system, which not only simplifies the operation procedure but also reduces cross-contamination (Fig. 4A). Because heat-resistant Cas12b is compatible with the operating temperature (60 °C~65 °C) of loop-mediated isothermal amplification (LAMP), one-step detection can be realized. For example, HOLMESv2 (*Li et al., 2019*), created by combining heat-stable AacCas12b with LAMP, integrates nucleic acid amplification and target detection steps into a single system, enabling one-step detection, simplifying operation procedures and avoiding cross-contamination. In addition, the newly discovered *Brevibacillus sp.* Cas12b (BrCas12b) was shown to have superior heat resistance. The CRISPR single-pot assay for detecting emerging volatile organic compounds (VOCs) (CRISPR-SPADE), based on BrCas12b, can accurately detect SARS-CoV-2 variants *via* simple and robust one-pot detection with a sensitivity of 95% (Ct value ≤ 32) and above (*Nguyen et al., 2022*). Compared with other one-pot methods, the thermal stability of BrCas12b alleviates some limitations of LAMP primer design, allowing for greater flexibility in primer selection. However, simply combining nucleic acid amplification with the

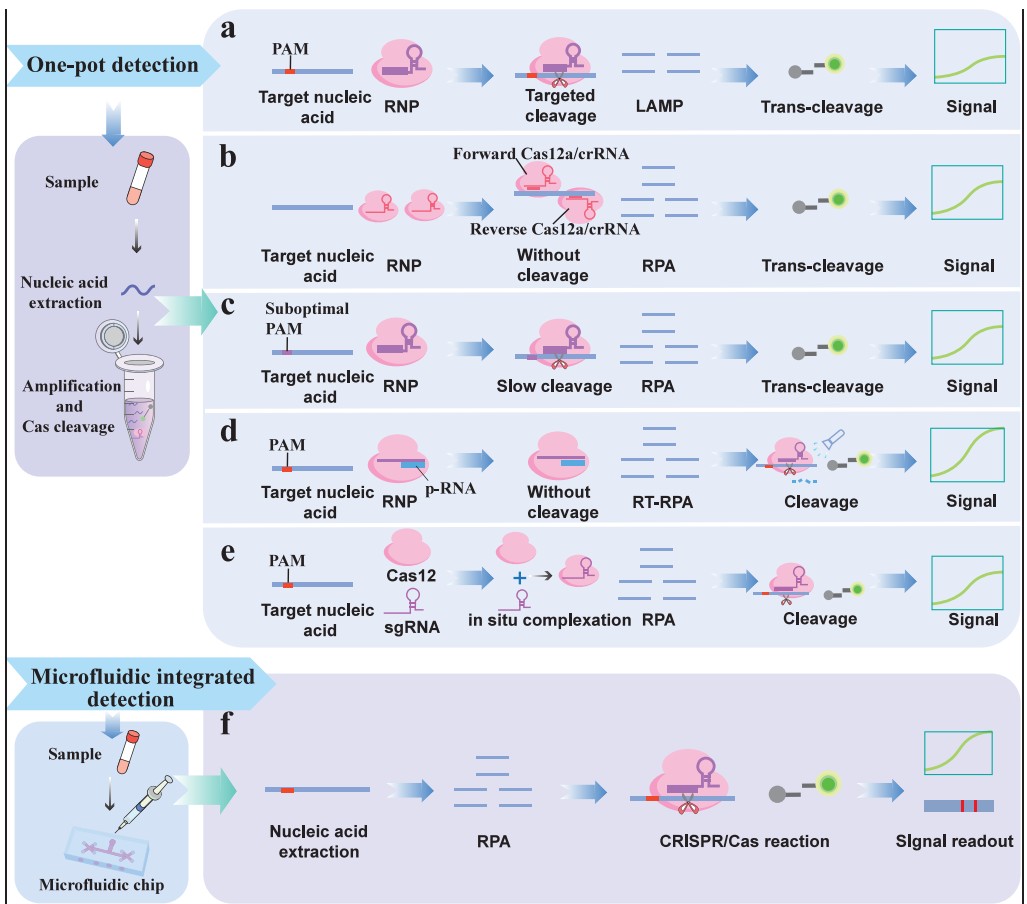

**Figure 4** **Integrated reaction detection strategy: (A) Traditional one-pot detection; (B) One-pot detection that does not rely on the PAM; (C) One-pot detection based on a suboptimal PAM; (D) Light-controlled one-pot detection; (E) *In situ* on-pot detection; (F) Microfluidic integrated detection.**

CRISPR/Cas system significantly affects detection sensitivity due to the competitive reaction between Cas-mediated *cis*-cleavage and target amplification. During target amplification, the target and primer are likely to be degraded by Cas-mediated *cis*-cleavage, which affects amplification efficiency (*Ali et al., 2020*; *Nguyen et al., 2022*).

Purposefully limiting the *cis*-cleavage rate of the Cas protein is a feasible strategy to improve the performance of one-pot assays. Therefore, to avoid the influence of *cis*-cleavage on target amplification, one study addressed this issue by employing an all-in-one double CRISPR/Cas12a (AIOD-CRISPR) detection method (*Ding et al., 2020*). The platform employs two PAM motif-free gRNAs that allow Cas12a-gRNA complexes (RNPs) to bind to amplification targets and induce Cas12a *trans*-cleavage activity to cleave quenched ssDNA probes but does not induce *cis*-cleavage activity, thereby preventing cleavage of the bound amplification target and attenuation of the amplification reaction (Fig. 4B). In addition, the suboptimal PAM of Cas12a (sPAMC)-based test, which was developed using gRNA targeting suboptimal PAM sequences, can also effectively inhibit *cis*-cleavage of Cas proteins and improve the sensitivity of one-pot assays (*Lu et al., 2022*). Cas12a reactions

based on suboptimal PAM exhibit weaker and slower *cis*-cleavage activity than reactions involving standard PAM, thus effectively slowing the degradation of targets and primers by the CRISPR/Cas system during nucleic acid amplification, allowing the amplification to accumulate sufficient products for subsequent CRISPR/Cas detection (Fig. 4C). However, the Cas-mediated *cis*-cleavage effect of the above methods still exists.

A light-controlled one-pot assay can better regulate crRNA-guided Cas enzyme activity in both spatial and temporal dimensions, effectively addressing the influence of *cis*-cleavage on target amplification efficiency. For example, one study designed a photocleavable protective RNA (p-RNA) to block the activity of crRNA (Fig. 4D) (*Hu et al., 2022*). After preamplification, p-RNA was degraded and isolated from the crRNA under light irradiation to restore Cas12a activity. This new one-pot detection system separates target amplification from CRISPR detection in space and time, thus eliminating the influence of CRISPR/Cas cleavage on the amplification reaction and achieving a sensitivity comparable to that of PCR. However, this method requires the synthesis of customized crRNA and additional human intervention, which hinders its simplification and automated detection. Recently, a study proposed an *in situ* one-pot assay that only achieved a sensitive one-pot assay by utilizing the slow association kinetics between the target-specific sgRNA and Cas12a to limit the cis exonuclease activity of Cas12 (Fig. 4E) (*Lesinski et al., 2024*). In the early stages of the reaction, RNP concentrations are limited by relatively slow kinetics, thereby circumventing the disadvantages associated with early-reaction cis exonuclease activity and retaining the signal transduction benefits of high RNP concentrations. However, owing to the overall decrease in RNP concentration, the *trans*-cleavage of the ssDNA reporter is reduced, which affects the final detection sensitivity.

### Microfluidic integrated detection

Microfluidic chips typically integrate miniaturized flow channels, valves, reaction chambers and detectors, enabling the integration of nucleic acid extraction, amplification, and CRISPR/Cas detection on a minimal chip (*Li et al., 2023c*). Once a drop of sample is added to the microfluidic chip, it automatically flows through all the components in the chip and eventually generates signals, eliminating the need for manual pipetting and enabling fast, high-throughput, multichannel, and digital detection (Fig. 4F) (*Chen et al., 2023b*). *Lu et al. (2023)* combined digital microfluidic technology with CRISPR/Cas12a to develop a fluorescence readout platform for detecting *Staphylococcus aureus* (*S. aureus*). *Staphylococcus aureus* cells were enriched from the raw materials *via* immunomagnetic beads. The enriched magnetic beads were collected by centrifugation, and the super-serum containing *S. aureus* cells was added to the digital microfluidic chip for further analysis. Cell lysis, RPA amplification, and Cas12a *trans*-cleavage occur automatically in sequence on the digital microfluidic chip. Due to its small volume and automation capabilities, the platform can complete detection in 55 min with a LOD of 32 CFU/mL. *Li et al. (2022c)* developed a simple, sensitive, and instrument-free CRISPR-based method for diagnosing SARS-CoV-2 using an independent microfluidic system. The platform utilizes a low-cost, portable hand warmer to culture microfluidic chips, maintaining optimal temperatures for RPA amplification and *trans*-cleavage. The microfluidic chip integrates isothermal amplification,

CRISPR cleavage, and lateral chromatography detection in a closed microfluidic platform, enabling pollution-free visual inspection that can detect up to 100 copies of SARS-CoV-2 RNA.

In addition, *Zhang et al. (2024a)* designed a multiplex, portable, centrifugal microfluidic detection system that integrates magnetic bead-based nucleic acid extraction, RPA amplification, and CRISPR-Cas13a detection in a single user step, with an overall turnaround time of 45 min. All reagents are preloaded onto the chip and can be automatically released. By utilizing a multichannel chip, it can simultaneously detect 10 infectious viruses with a LOD of 1 copy per reaction. The self-contained reaction system, designed using a microfluidic chip, can effectively prevent contamination and potentially achieve the goal of being pipette-free. Additionally, well-designed microchannels can distribute a single sample across multiple reaction chambers to achieve multiple detections (*Ackerman et al., 2020*; *Shang et al., 2024*). However, the complex preparation of microfluidic chips limits their widespread use, which requires further commercialization and the integration of instruments and chips to facilitate POC testing.

## Portable output strategy

Most CRISPR/Cas pathogen detection methods utilize quenched or fluorophore-modified ssDNA probes to generate spectrally readable fluorescence signals. However, owing to the dependence on complex fluorescence detection equipment, the sensitivity and portability of these readout strategies still have much room for improvement. Therefore, several portable readout strategies are used for CRISPR/Cas detection.

### *Ultrasensitive lateral flow chromatography system*

The lateral flow assay (LFA) is a portable, easy-to-operate, and cost-effective solution for resource-poor areas and POCT. The principle is that the DNA probe sequence complementary to the reporter is fixed on the test line of the lateral flow rod to capture the reporter. When the target is present, the reporter is cleaved, resulting in the generation of a light signal on the test line. Due to their unique advantages of simplicity and portability, LFAs, such as SHERLOCK v2 (Table 2) (*Gootenberg et al., 2018*) and STOPCovid (*Joung et al., 2020*), have been widely utilized in CRISPR assays; however, their sensitivity is generally low. Therefore, to improve the sensitivity of CRISPR/Cas-LFA-based assays, *Zhou et al. (2022)* utilized quantum dot-enhanced fluorescence signals to develop a fluorescence-enhanced LFB based on CRISPR/Cas12a, known as CRISPR/Cas-recombinase-assisted amplification-based LFB (CRA-LFB), for detecting *Staphylococcus aureus*. QDs are fluorescent semiconductor nanocrystals with high brightness, wide excitation and strong light stability (*Deng et al., 2017*). On this platform, streptavidin (SA) modified with quantum dots (QDs) is embedded on the T-line, allowing for the generation of a large number of specific amplification products through recombinant enzyme-assisted amplification (RAA). The *trans*-cleavage activity of Cas12a is then activated to degrade the biotinylated probes. The degraded probe cannot be captured by the QD-SA group on the T-line, resulting in no signal accumulation. In contrast, a significant fluorescence signal is observed on the C-line, and the target concentration is negatively correlated with the

fluorescence intensity on the T-line. The limit of detection for *Staphylococcus aureus* DNA was 75 aM.

Similarly, *Zhong et al. (2023)* utilized the fluorescence enhancement characteristics of QDs to develop a CRISPR/Cas12a-based QD nanobead (QDNB)-labelled lateral flow assay (CQ-LFA) for detecting varicella zoster virus (VZV) with results that were completely consistent with those of q-PCR. The platform introduced the QDNB-anti-FAM antibody conjugate into the Cas12a cleavage system. When the target DNA was present, the cleavage activity of Cas12a was triggered to degrade the biotin-ssDNA-FAM probe, and the QDNBs leaked out of the streptavidin-embedded T-line, resulting in no or a weak fluorescence signal on the T-line. Instead, in the absence of a target, the reporter is intact and bound to the T-line, where a fluorescent signal is visible.

### Smartphone readout system

Compared to the naked eye, smartphones are capable of converting the signals generated by CRISPR/Cas cleavage into accurate digital data, displaying more sensitive and precise readings. They offer the advantages of low cost, wide availability, and high portability. In addition, smartphones also have great potential and broad application prospects in statistics, summary and query data. *Fozouni et al. (2021)* designed a smartphone-based fluorescence microscope and reaction chamber to quantify the fluorescence signal produced by Cas13a, enabling direct detection of SARS-CoV-2. Detection integrated with a mobile phone-based reader device is expected to achieve rapid, low-cost, quantitative, immediate screening of SARS-CoV-2. *Yin et al. (2021)* developed a G-quadruplex-based CRISPR/Cas12a bioassay with smartphone readings for pathogen detection. On this platform, ssDNA is designed with a guanine-rich sequence, and the addition of $K^+$ forms a stable G-quadruplex DNase. DNase can catalyze the TMB-H2O2 reaction in the presence of heme, resulting in changes in absorbance and color at 454 nm, which can be easily distinguished by the naked eye and smartphones using a color selecting app. The LOD of this strategy for *Salmonella* was 1 colony-forming unit (CFU) per millilitre (mL). In addition, *Song et al., (2022)* reported a colorimetric DNA enzyme reaction triggered by LAMP and CRISPR/Cas9, and developed a machine learning (ML)-based smartphone application to check the diagnostic results of SARS-CoV-2 and variant strains periodically. The CRISPR/Cas9 system eliminates false-positive signals generated by LAMP, thereby improving the accuracy of detection. Smartphones achieve end-user-friendly detection. The system can detect SARS-CoV-2 and its variants quickly and accurately within 1 h in the field, with sensitivity reaching the attomolar level.

### Innovative small readout system

The combination of various innovative small readout strategies, such as fluorometers, thermometers, and glucose meters, with the CRISPR/Cas system, can achieve portable and sensitive POC diagnosis of pathogens, offering the advantages of low cost and user-friendliness. *He et al. (2020)* designed and developed a fluorescence-based POC system for detecting African swine fever virus (ASFV). The POC system consists of a disposable filter element that can hold up to 80 individual samples and a custom fluorometer. In the presence of ASFV DNA, the Cas12a/crRNA complex was activated and degraded the fluorescent

ssDNA reporter. The system can detect ASFV at concentrations as low as 1 pM within 2 h without nucleic acid amplification. *Liu et al. (2022)* utilized a handheld thermometer as a readout tool to develop a CRISPR/Cas12a-based photothermal platform for detecting citrus-related *Alternaria* genes. The AuNPs modified with HRP were linked to MB through ssDNA. HRP can oxidize TMB to form oxTMB, which not only appears blue but also exhibits a strong photothermal effect driven by a near-infrared laser. A laser can irradiate it, producing a significant temperature increase, and the changes can be easily recorded with a handheld thermometer. The platform exhibits high sensitivity for semiquantitative detection, with a LOD of 1.5 pM. Recently, *Li et al. (2023b)* combined a microfluidic platform with glucose biosensing technology to develop a bioinspired CRISPR-mediated cascade reaction biosensor (CRISPR-MCR) for HIV point-of-care diagnosis. HIV nucleic acid can be digitally detected by a personal blood glucose meter at the endpoint, similar to reading glucose levels in the blood. The detection sensitivities of HIV DNA and HIV RNA *via* the platform are 43 copies and 200 copies per test, respectively, without the requirement for expensive instruments.

The portable readout strategy enables accurate and sensitive signal readout with minimal instrumentation, demonstrating great potential for POC detection. However, most strategies still require complex sample extraction, preamplification and pipetting steps, making the overall strategy unsuitable for field point-of-care diagnosis. In future work, an extraction-free and amplification-free integrated reaction strategy will be combined with a portable readout to achieve real-time, on-the-spot diagnosis.

## OTHER APPLICATIONS OF CRISPR-BASED DIAGNOSTICS

The detection of cancer-associated biomarkers is beneficial to the prognosis of cancer patients (*Liu et al., 2018*). The application of CRISPR technology in cancer diagnostics is rapidly expanding, and its high sensitivity, programmability, and rapid detection capability provide a revolutionary tool for the early diagnosis of cancer biomarkers. For example, CRISPR technology is used for the rapid detection of circulating tumour DNA (ctDNA) and microRNA (miRNA) in human circulation. The Cas12a detection system for epidermal growth factor (EGFR) mutation detection can be used to assess cancer progression (*Tsou, Leng & Jiang, 2020*). It can detect EGFR858 and EGFR790 within less than 3 h with a LOD of 0.005%. Currently, a direct detection method for Cas13a-mediated miRNAs has been developed, which can detect the target miRNA within 30 minutes with a detection limit of 4.5 aM (*Shan et al., 2019*). This method was employed for the relative quantitative detection of miR-17 in four cell lines, yielding results consistent with those obtained by qRT-PCR. Single nucleotide polymorphisms (SNPs), as key markers of cancer genetics and molecular stratification, play a crucial role in the early detection, prevention and treatment of cancer. The high sensitivity, specificity and portability of the CRISPR assay platform are promoting the development of precise SNP diagnosis in a faster and more portable direction. The SHERLOCK system has been used to differentiate between the Zika virus (ZIKV) and dengue virus (DENV) subtypes (*Myhrvold et al., 2018*). The Cas14a-based DETECTR platform has also been utilized to identify single nucleotide mutations in HERC2 (*Harrington et al., 2018*).

In addition, CRISPR/Cas detection technology has gradually expanded from nucleic acid detection to the detection of proteins (*Dai et al., 2019*), small molecules (*Liang et al., 2019*; *Shu et al., 2022*), and cells (*Shen et al., 2020*). Since the CRISPR/Cas system itself cannot recognize these nonnucleic acid targets, the detection of these substances is mainly mediated by proteins or aptamers (ssDNA or ssRNA) that undergo conformational changes upon target recognition. Due to the established protocols of the CRISPR-Cas system in living cells, a CRISPR-mediated imaging strategy can be employed for real-time genome tracking, offering a potential solution for verifying the accuracy of gene editing (*Wang et al., 2019*).

## CONCLUSIONS AND PERSPECTIVES

In the past few years, significant progress has been made in the application of CRISPR/Cas technology due to its advantages in economy, speed, sensitivity, and specificity. Various simplified detection strategies have been proposed to further promote the clinical application of CRISPR/Cas technology, especially for large-scale diagnosis and field detection of pathogens in remote areas. However, to truly transform these findings from preclinical diagnostic tests to clinical applications, many obstacles still need to be overcome in the future.

### Challenges in clinical application

Currently, the sensitivity of most CRISPR/Cas assays still needs to be considered in relation gold-standard PCR, especially for portable readout strategies, which often fail to achieve ultrasensitive detection. Therefore, more sensitive methods are needed to improve the detection performance of CRISPR/Cas assays, which can be achieved by selecting better effector components (including Cas protein, gRNA, and highly sensitive reporter probes) (*Chen et al., 2024b*), adjusting the ratio of Cas to gRNA (*Guo et al., 2020*; *Joung et al., 2020*), modifying the structure of gRNA to increase the *trans*-cleavage activity of the Cas protein (*Nguyen, Smith & Jain, 2020*), and optimizing buffer conditions or adding chemical additives (*Joung et al., 2020*; *Ma et al., 2020*). Additionally, the standardized detection of CRISPR/Cas technology is crucial for promoting its clinical application. Different experimental environments (temperature, pH, and ion concentration) (*Dai et al., 2019*) and operating procedures (specimen sampling, nucleic acid extraction, data processing, and reporting methods) may affect the final detection results, making the readout results unstable, which hinders their transformation into a usable diagnosis for end-users.

Although CRISPR/Cas technology has broad prospects in the fields of gene editing and diagnosis, its off-target effects and ethical concerns have consistently been the primary obstacles to its clinical application. Off-target effects may lead to unpredictable consequences, such as sequence mutations, deletions, rearrangements, immune responses and oncogene activation, which limits the application of CRISPR technology in clinical therapy (*Chen et al., 2020*). Studies have shown that shortening the crRNA spacing region or modifying the direct repetition of crRNA leads to stricter specificity, thereby reducing the occurrence of off-target effects (*Vargas et al., 2024*). The ethical and safety issues

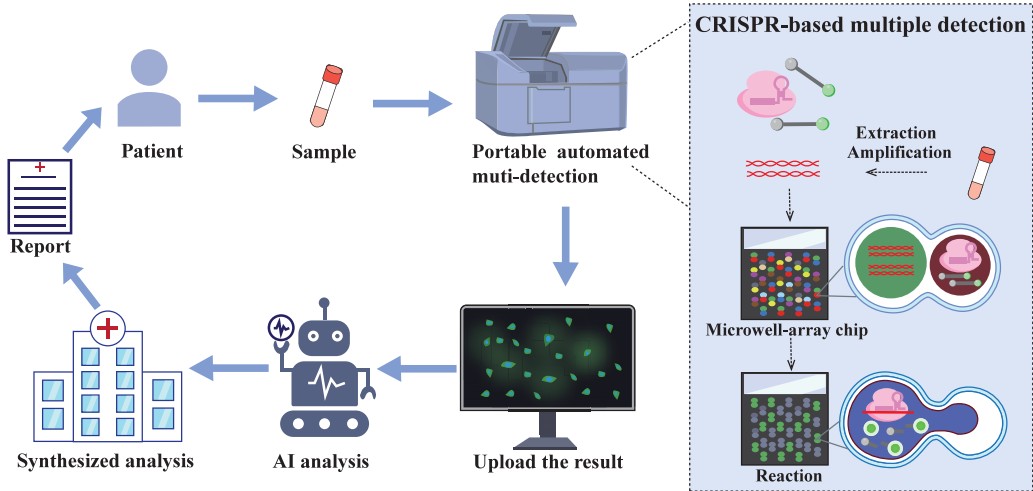

**Figure 5** **The future of CRISPR/Cas automated multi-detection system.** Samples collected from patients in communities and care points can be tested cheaply and quickly through portable integrated testing equipment. The integrated equipment combined with droplet microfluidic chip technology enables a single sample to simultaneously detect multiple targets. The test results will be uploaded *via* the network, screened by artificial intelligence (AI), and finally a comprehensive analysis will be conducted to obtain a test report, which will provide medical guidance for patients.

surrounding CRISPR technology are of great concern worldwide. As the technology is still in its infancy, significant work remains to be done to improve its accuracy and ensure that alterations to the genome do not lead to unforeseen consequences, particularly when applied to human trials.

## Technological advancements and strategic orientation

In recent years, the development of quantitative, multiplex, and integrated CRISPR assays has garnered significant attention. Currently, digital droplet and droplet microfluidic technologies show great potential for quantitative and multiplexed detection (*Roh et al., 2023*). CARMEM based on droplet microfluidics can achieve large-scale pathogen screening and improve sequencing efficiency, which is crucial for comprehensive surveillance and epidemiological research of any epidemic. Additionally, CRISPR technology has currently realized one-pot multipathogen detection (*Shang et al., 2024*). However, with the continuous popularization of automation, the automation of CRISPR diagnostic technology is almost inevitable. This automation of testing reduces the difficulty of work and labour costs, accelerates detection speed, and unifies diagnostic criteria (*Tozzoli et al., 2015*). In addition, the field of POC diagnosis has been the primary clinically oriented method for CRISPR diagnosis. Therefore, the development of automated multiplexed pathogen detection devices that allow the detection of a variety of pathogen targets in a single reaction system will be emphasized in future efforts (Fig. 5). The main purpose is to integrate CRISPR/Cas technology with droplet microfluidic technology and artificial intelligence so that sample processing, preamplification, the CRISPR reaction and droplet pairing, as well as readout into a single device, achieve one-step detection of multiple

pathogens in a single sample and monitoring of pathogen drug resistance mutations. The integrated components include not only the completion of reactions but also the digital display of results, as well as the online uploading and analysis of the diagnostic results. Designing a fully integrated CRISPR assay device presents significant technical challenges, which necessitate continuous improvement of each component to achieve optimal assay performance.

The list of abbreviations is provided in Table S2.

### Funding
This work was supported by the National Natural Science Foundation of China (3210119), the Major Scientific and Technological Innovation In Chengdu In 2021 (2021-YF08-00119-GX), the Xinglin Scholar Research Premotion Project of Chengdu University of TCM (ZYTS2023014), and the Sichuan Provincial Regional Innovation Cooperation Project (No. 2023YFQ0075). The funders had no role in study design, data collection and analysis, decision to publish, or preparation of the manuscript.

### Grant Disclosures
The following grant information was disclosed by the authors:
National Natural Science Foundation of China: 3210119.
Major Scientific and Technological Innovation In Chengdu In 2021: 2021-YF08-00119-GX.
Xinglin Scholar Research Premotion Project of Chengdu University of TCM: ZYTS2023014.
Sichuan Provincial Regional Innovation Cooperation Project: 2023YFQ0075.

### Competing Interests
The authors declare there are no competing interests.

### Author Contributions
- Yan Wu conceived and designed the experiments, performed the experiments, performed the computation work, prepared figures and/or tables, authored or reviewed drafts of the article, and approved the final draft.
- Jimin Li conceived and designed the experiments, performed the experiments, performed the computation work, authored or reviewed drafts of the article, and approved the final draft.
- Rui Wang conceived and designed the experiments, performed the experiments, authored or reviewed drafts of the article, and approved the final draft.
- Fengling Qiao conceived and designed the experiments, analyzed the data, authored or reviewed drafts of the article, and approved the final draft.
- Jinlin Guo conceived and designed the experiments, analyzed the data, authored or reviewed drafts of the article, and approved the final draft.
- Xu Jia conceived and designed the experiments, analyzed the data, authored or reviewed drafts of the article, and approved the final draft.

## Data Availability

This is a literature review.

## Supplemental Information

Supplemental information for this article can be found online at http://dx.doi.org/10.7717/peerj-achem.36#supplemental-information.

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
