# Peer review of "Progress in applying CRISPR in pathogen detection"

_PeerJ Analytical Chemistry, doi:10.7717/peerj-achem.36_

## Round 0.1 · original submission · Major Revisions

The reviewers collectively recommend enhancing the manuscript’s clarity, structure, and rigor by addressing several key areas. First, they emphasize the need to refine the focus on CRISPR-based pathogen diagnostics in the title, abstract, and text while distinguishing between gene editing and nucleic acid detection applications. Structural improvements should include a methodology section outlining research questions, inclusion/exclusion criteria, a clear scope, and a comparison with existing reviews to highlight novelty and justify the review’s necessity. Content revisions should expand discussions on Cas14, ethical challenges, and DNA repair mechanisms (NHEJ/HDR) while correcting inaccuracies (e.g., Cas13’s functionality) and clarifying technical points (e.g., mismatch tolerance and PAM annotations). Figures and tables require refinement for accuracy, detail, and educational value, including corrected schematics, additional visuals summarizing CRISPR strategies, and comparative tables of detection tools. Terminology consistency, complete definitions for abbreviations (e.g., PAM, LOD), and language polishing are essential to improve readability. The conclusions should focus on CRISPR’s diagnostic advancements rather than generic criteria. Finally, the authors should articulate the review’s novelty, address the literature selection methodology, and conduct thorough language editing to resolve grammatical errors and awkward phrasing, ensuring the manuscript meets high academic standards.

**Language Note:** The review process has identified that the English language must be improved. PeerJ can provide language editing services - please contact us at [email protected] for pricing (be sure to provide your manuscript number and title). Alternatively, you should make your own arrangements to improve the language quality and provide details in your response letter. – PeerJ Staff

·

Basic reporting

-

Experimental design

-

Validity of the findings

-

Additional comments

1. I will suggest providing a comparison with related reviews to establish the limitations of existing reviews and the need for this review.
2. A methodology section is needed to highlight research questions, inclusion, and exclusion criteria.
3. Authors can also add a scope of the review.
4. Cas13 is mentioned, but what about Cas14?
5. Section 3 should explain Nonhomologous end joining and Homologous Directed Repair
6. Ethical regulations must be discussed as part of the challenges of the CRISPR/Cas system
7. There should be a link between one section to another to improve flow.
8. What is the principal finding of the research?
9. What are the practical implications of the review?
10. Authors can add a list of abbreviations
11. A figure can be added to summarize the strategies of CRISPR CA's system for NA detection.

Cite this review as

·

Basic reporting

-

Experimental design

-

Validity of the findings

-

Additional comments

This review begins by introducing the shortcomings of traditional pathogen detection methods, providing a foundation for the subsequent discussion of the CRISPR/Cas system and its distinctive characteristics. Specifically, the properties of Cas9, Cas12, and Cas13 are being evaluated in detail. The review then primarily focuses on the simplified detection strategies, such as nucleic acid extraction-free, amplification-free, integrated reaction, and portable output strategies, which are discussed in detail. In the concluding section, the review highlights the current problems that need to be improved, providing direction for future research. This review makes a significant contribution to the field of CRISPR-based detection technologies for pathogenic bacteria, with its practical relevance, well-structured framework, and rigorous data collection, providing valuable insights for researchers. There are a few points in the manuscript that could be further clarified and strengthened. More detailed comments are as follows:

Major Comments:
1. CRISPR technology was initially developed for gene editing. It is worth noting that in recent years, a series of nucleic acid detection technologies based on the CRISPR system have also emerged. “Gene editing” and “nucleic acid detection” are two important yet distinct applications of the CRISPR system. Given that this review primarily focuses on CRISPR-based pathogen diagnostics, it might be appropriate to refine the title and certain sections of the text to more clearly emphasize this focus. The adjustments in this direction could further strengthen the clarity and precision of the work.

2. The abstract could benefit from more precise wording. For instance, the challenges mentioned in lines 36-37, such as cumbersome operation, cross-contamination, and reliance on complex instruments, seem to arise more from the integrated technologies or experimental workflows rather than being inherent limitations of the CRISPR system itself. Clarifying this distinction could help improve the accuracy of the description.

3. Some sections of the article could use more precise wording to ensure clarity, particularly for readers who may be less familiar with the subject. In the introduction, it is necessary to take CRISPR technology itself as a starting point, deeply analyze its principles, advantages, and limitations. On this basis, further explore how to optimize its application in the field of pathogen detection by integrating other strategies and technologies. Additionally, distinguishing between the “CRISPR system” and “CRISPR-derived nucleic acid detection technologies” would improve clarity. For instance, the content in Lines 79-83 could be refined to avoid potential misunderstandings.
Furthermore, it might be helpful to provide a more detailed explanation of how CRISPR-based strategies and technologies address the limitations of traditional methods. Rather than simply listing technical approaches (lines 84-97), highlighting the specific improvements and their practical applications would make the content more accessible and convincing for readers. A careful revision of these sections could enhance clarity and reliability.

4. In the introduction of Type VI (Cas13), the statement that “Cas13 is the only class 2 ribonuclease family that targets RNA, preventing permanent damage to an organism’s DNA” may need revision, as it appears to be inaccurate (lines 174-175 and table 1). Additionally, recent research has revealed a significant breakthrough in CRISPR-based diagnostics: the discovery of LbuCas13a’s ability to directly target DNA without the restrictions of Protospacer Flanking Sequence (PFS) and Protospacer Adjacent Motif (PAM) sequences, combined with its robust trans-cleavage activity, which could be a valuable addition to the discussion.

5. The content description in “...the more mismatch-tolerant Cas proteins, precise recognition of target genes can be achieved.” might need some re-evaluation (line 197). Why can precise target identification be achieved if mismatch tolerance is higher? If the intention is to highlight that introducing mismatches in crRNA improves the detection of single-base mutations, the wording could be adjusted to better reflect this idea.

6. While the article is written in professional English and generally adheres to academic standards, some sentences could benefit from clearer logic and more precise wording and language polish work (e.g., lines 192-193, 310, 421-423, 425, 435, 578, 620, 666, 683, etc.). A thorough review to refine these sections and improve the flow with more appropriate connectives would enhance readability and ensure the information is accurately conveyed for international readers.

7. Figure 3 is well-designed overall, but there are a few minor graphical details that could be refined for greater accuracy. In Figure 3a, the arrow direction seems to be labeled incorrectly, and the cleavage site in the scgRNA might be more accurately represented as a partially open-loop structure rather than showing both an open loop and a complementary strand at the same time. In Figure 3b, it might be clearer to remove the target nucleic acid from the detection reaction system, as the distinction between positive detection (with target) and negative detection (without target) can be demonstrated in the downstream figure panels when the system does not contain the target.

8. Figure 4 effectively presents integrated reaction detection strategies, but a few adjustments could improve its clarity. For instance, in Figure 4a, it would be clearer to replace the amplification with specific amplification methods. LAMP might be a more suitable choice, as RPA amplification would make it difficult to achieve effective target detection in the one-pot assay without any auxiliary strategies. Additionally, the description in Figure 4b could be clarified. A simple schematic of dual crRNAs sites might cause confusion for readers unfamiliar with the field, leading them to confuse that dual-crRNA in general facilitates the cis-cleavage activity of Cas, but why does it favor the one-pot assay? To address this, it would be more accurate to explicitly illustrate that the two crRNA sites are located downstream of the forward and reverse primer binding sites, respectively.

9. Please note the positions of the PAM site labeled in the target sequence in Figure 1, Figure 3, and Figure 4. The PAM sequences are critical for Cas protein recognition and binding, and their accurate annotation is essential for understanding the targeting mechanism and designing effective guide RNAs (gRNAs).

10. The “Conclusions and Prospects” section provides a good overview, but it could be more focused and concise. Since many CRISPR-based detection technologies with high sensitivity and specificity have already been developed, it would be valuable to focus the discussion on the progress and application achievements of CRISPR-based pathogen detection methods rather than broadly summarizing the ideal detection criteria, such as sensitivity and specificity. This refinement would help to highlight the unique contributions and potential of CRISPR-based pathogen detection.

General comments:
1. “PAM” should be “Protospacer Adjacent Motif (PAM)”, “RPA” should be “Recombinase Polymerase Amplification (RPA)”, “POCT” should be “Point-of-Care Testing (POCT)”, “POC” should be “Point-of-Care (POC)” and other abbreviations should be provided with the full terms along with their abbreviations to improve clarity and make the content more accessible to all readers.
2. “The detection limit of...” should be corrected to “Limit of Detection (LOD)” for more accuracy.
3. “...In addition to precise dsDNA cleavage (cis-cleavage), Cas12a also shows trans-cleavage activity. When crRNA attaches to complementary ssDNA, Cas12a cleaves any ssDNA molecule into single-nucleotide/dinucleotide” should change into “Target DNA”. Both ssDNA and dsDNA can activate the trans-cleavage activity of Cas12a.
4. “...are limited to culturable batteries” should be a typo (line 66). The word should be changed to “bacteria”.

Cite this review as

Reviewer 3 ·

Basic reporting

-

Experimental design

-

Validity of the findings

-

Additional comments

Summary

Yan Wu and co-authors describe and summarize various applications of CRISPR-based gene editing technologies for pathogen detection, with a particular focus on strategies aimed at simplifying diagnostic workflows for real-world clinical and field applications. It begins by outlining the limitations of conventional diagnostic methods, such as culture, immunoassays, and PCR, and positions CRISPR as a transformative alternative due to its programmability, specificity, and rapid turnaround time. The authors detail the molecular characteristics and diagnostic potential of key CRISPR effector proteins—Cas9, Cas12, and Cas13—highlighting their cleavage activities and target specificity. The core of the review is devoted to emerging strategies designed to overcome the practical limitations of current CRISPR diagnostics, categorized into four themes: nucleic acid extraction-free workflows, amplification-free detection systems, integrated one-pot reactions, and portable output platforms. Each strategy is illustrated through various platforms (e.g., SHERLOCK, DETECTR, STOPCOVID, FIND-IT) and is supported by examples of innovations in sample handling, signal amplification, and readout formats, such as lateral flow assays, smartphone interfaces, and electrochemical biosensors. The review concludes by identifying current bottlenecks—such as insufficient sensitivity in some amplification-free methods, lack of standardization, and challenges in achieving quantitative and multiplex detection—and suggests future directions to bridge the gap between preclinical innovation and widespread clinical deployment.

General comments

This manuscript is well-structured, topical, and comprehensive in scope, offering a relevant and timely overview of the progress in CRISPR gene editing technology for pathogen detection. The authors do a commendable job of covering a wide range of CRISPR systems and diagnostic platforms, with particular emphasis on simplified and field-deployable detection strategies. The manuscript balances technical detail with conceptual clarity, making it informative for both newcomers and specialists in the field. The descriptions of cleavage mechanisms, diagnostic workflows, and Cas protein functionalities are generally accurate and supported by a solid body of literature, with a broad and representative selection of references.

However, there are several areas where the manuscript would benefit from improvement. While the text is informative, the figures and tables do not yet match the quality of the narrative.

Specifically, the visual elements are underdeveloped and overly simple, and lack explanatory detail or comparative clarity. Incorporating additional figures that cover different sections of the review article, along with summary tables (summarizing detection tools, platform features, shortcomings, and comparison metrics (e.g., sensitivity, specificity, output type)) would greatly enhance the didactic value of this review. Furthermore, although the article is positioned as a comprehensive review, the methodology for literature selection is vague, and the novelty or distinct contribution of the review is not sufficiently articulated in contrast to existing reviews in the field. There are also linguistic and stylistic issues throughout the manuscript, including awkward phrasing, inconsistent terminology, and a number of grammatical and typographical errors, which require careful revision and a professional language check. These and other concerns are detailed below, and I encourage the authors to address them to bring the manuscript up to the standard of a high-quality, instructive review article.

Strengths

• Clear and explicit content
• Comprehensive structure, in the general organization of the manuscript, and also, per (sub)section
• Extensive citation of relevant publications in the field
• Extensive and instructive examples throughout the manuscript.

Minor Weaknesses:
• The review’s methodology is underdeveloped and lacks the transparency needed to validate its comprehensiveness. On Page 2, Lines 53-59, the authors merely state that “a large number of documents” were searched on PubMed through the Internet, without specifying the date range, number of included articles, or the selection criteria beyond the presence of keywords. This vagueness undermines the credibility of the review and limits its reproducibility. Even if the paper is a narrative review, it should incorporate basic transparency elements-such as a flowchart of selection, exclusion criteria, or the approximate number of studies evaluated strengthen its foundation.

• The manuscript contains multiple awkward phrasings, informal metaphors, and typographical errors that detract from its scientific professionalism. For example: Page 1, Line 28: The phrase “powerful molecular scissors” is colloquial and oversimplified for a scholarly article. A more suitable phrase would be “programmable endonuclease-based gene editing system.” Page 2, Lines 49-50: The phrase “new ideas... and new methods” is redundant and could be revised to “innovative strategies for pathogen detection and improved clinical applicability.” Page 3, Line 66: A critical typo appears with the term “culturable batteries,” which should clearly be “culturable bacteria.” In addition, some sentences are unnecessarily long and convoluted, reducing readability. The manuscript would benefit significantly from a careful linguistic revision by a native English speaker or a professional editing service.

• Although the review does a good job of categorizing various CRISPR-based diagnostic tools, it largely avoids engaging in evaluative comparison or critique. This is a missed opportunity. Readers would benefit from a more systematic comparison of technologies across parameters such as: Target molecule (RNA vs DNA); Detection limits and speed; Requirements for amplification; Suitability for field vs clinical lab settings; Cost and scalability

• The summary Table 2 at the end of the manuscript is a valuable inclusion, but is not referenced or integrated into the main text. For example, when describing DETECTR, SHERLOCK, or STOPCOVID, the authors should point to the corresponding rows in the table to facilitate cross-reference. Better yet, condensed comparative sub-tables within each major section could enhance clarity and accessibility.

• There are instances of unnecessary repetition across sections. For instance, the content on Page 5, Lines 104-108, closely mirrors the abstract's final lines, offering no new synthesis or transition. Additionally, sections such as “Target Audience” (Page 2, Line 47) feel out of place in a scientific review and could be omitted or merged into the introduction without loss of content.

• The concluding section (Page 26, Lines 691-741) identifies several key themes, such as the need for higher sensitivity, specificity, and standardized protocols. However, the discussion remains quite general. The authors stop short of offering strategic insight-for example, which simplified detection strategy holds the most promise for near-term translation? What technical or regulatory barriers are the most pressing? A summary figure or roadmap laying out “next steps” would elevate the impact of the review considerably.

• Reducing the number of reported studies and adding more text to analyze the progress made in each direction is highly recommended. A review article should report the studies that made a significant contribution, with a focus on analyzing and presenting this contribution to the community.

• It will be interesting to see a tabled comparison (to detail cons vs. pros) that includes each of the described nano-based approaches and how each method can be useful to overcome the challenges in implementing CRISPR-based gene editing and engineering.

• How were these nanosystems successful in the delivery of other DNA drugs (e.g., DNA vaccines, plasmids, etc.)? How and why will they be better for CRISPR delivery? Readers will be interested to see such a high-level analytical point of view.

• The text needs an additional language check for grammatical errors and typos.

• The Figures: Almost all labels in the figures are too small to read (even for people with perfect vision). Many of the graphics in the figures are too small as well, or have poor contrast and are complex (too many details to follow in the same figure).
Specific weaknesses

• Needs a major effort for editing towards improving the writing flow and grammar.

Cite this review as
Anonymous Reviewer (2025) Peer Review #3 of "Progress in applying CRISPR in pathogen detection (v0.1)". PeerJ Analytical Chemistry

---

## Round 0.2 · accepted · Accept

Thanks for responding diligently to the editorial and reviewers' comments.

Reviewer 3 ·

Basic reporting

No further comments

Experimental design

No further comments

Validity of the findings

No further comments

Additional comments

No further comments

Cite this review as
Anonymous Reviewer (2025) Peer Review #3 of "Progress in applying CRISPR in pathogen detection (v0.2)". PeerJ Analytical Chemistry